# Bimodality of Sparse Autoencoder Features is Still There and Can Be Fixed

## Abstract

Sparse autoencoders (SAE) are a widely used method for decomposing LLM activations into a dictionary of interpretable features. We observe that this dictionary often exhibits a bimodal distribution, which can be leveraged to categorize features into two groups: those that are monosemantic and those that are artifacts of SAE training. The cluster of noninterpretable or polysemantic features undermines the purpose of sparse autoencoders and represents a waste of potential, akin to dead features. This phenomenon is prevalent across autoencoders utilizing both ReLU and alternative activation functions. We propose a novel training method to address this issue and demonstrate that this approach achieves improved results on several benchmarks from SAEBench. The code of the project can be found at https://anonymous.4open.science/r/sae_bimodality-ICLR.

## 1 Introduction

### 1.1 The Dawn of SAEs and the Forgotten Phenomenon

The superposition hypothesis (Elhage et al., 2022) posits that deep learning models represent more features than they have neurons through linear directions. In (Sharkey et al., 2023), the authors demonstrated that the simple architecture of sparse autoencoders (SAE) can perfectly "disentangle" superposition in a synthetic dataset. In this toy model scenario, we have access which features (directions in the activation space) are ground truth. The key challenge when applying SAEs to real language data lies in measuring a quality of features when there is no ground truth. The proposed method involves the following steps:

- Train a sparse autoencoder $A$ on a dictionary of size $N$.
- Train another sparse autoencoder $B$ on a dictionary of a larger size.
- Check what features are similar between the two by measuring **maximal cosine similarity (MCS)** between feature $i$ from $A$ and the entire dictionary from the larger dictionary $B$.

The rationale is that features identified independently by two dictionaries are, in a sense, universal." Following this procedure, for every feature $i$, a scalar score $MCS_i$ is obtained, which serves as a proxy for feature quality, monosemanticity, or universality. Surprisingly, it was soon observed that these scores follow a clear bimodal distribution (Cunningham & Riggs, 2023). Moreover, the lower values cluster closely with the distribution of randomly generated vectors (Huben, 2023). Subsequent manual and automatic approaches to interpretability (Bills et al., 2023) confirmed that the MCS value is positively correlated with feature monosemanticity. In (Riggs, 2023), the authors manually inspected features by sorting them in descending order based on the MCS score and discovered that the top-ranked features are monosemantic. Furthermore, (Cunningham, 2023) demonstrated that the top-150 MCS features exhibit higher interpretability scores compared to random ones.

### 1.2 A Simpler Feature Quality Proxy

The observation that MCS scores can be used to assess the quality of SAE features and distinguish between "true" directions and random artifacts appears to have been overlooked. This insight is notably absent in the main culmination paper (Huben et al., 2023) and is only briefly mentioned in a footnote in the independent Anthropic work (Bricken et al., 2023). We speculate that the primary

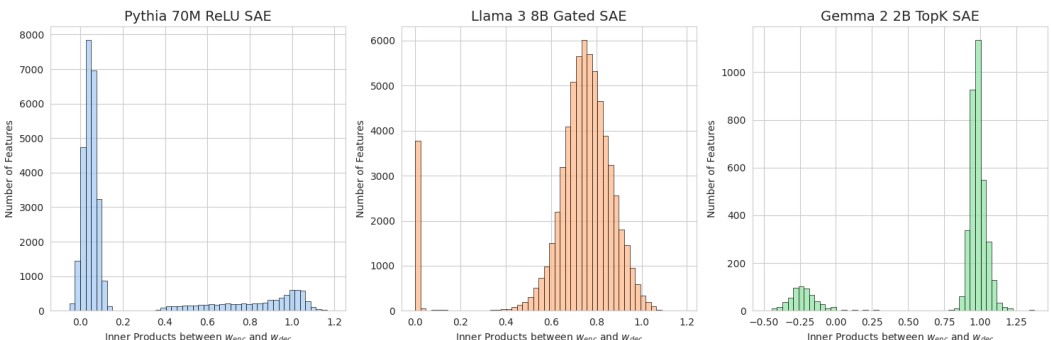

Figure 1: We can observe the bimodality of SAE features across different models and SAE architectures. The SAE details are described in B

reason for its neglect is the computational cost of the described method, as calculating MCS scores necessitates training an additional larger dictionary.

In our work, we revisit this idea by:

- Proposing a simpler scalar proxy for feature quality that does not require any data or additional training.
- Demonstrating that this score remains bimodal even in modern SAE variants, such as TopK, Gated, and JumpReLU.
- Establishing a positive correlation between the proposed score and the autointerpretability score.
- Proposing a novel training method to eliminate bimodality by design.
- Demonstrating that the proposed method outperforms standard training across multiple benchmarks, models, dictionary sizes, sparsities, and activation functions.

## 2 BACKGROUND ON SPARSE AUTOENCODERS

Let $\mathbb{R}^n$ represent the activation space of a deep learning model, such as the residual stream of a large language model (LLM). The goal of a sparse autoencoder (SAE) is to decompose the activation of a data point into a sparse linear combination of features from a dictionary of size $m$. It has been shown (Huben et al., 2023; Bricken et al., 2023) that such features are more interpretable than neuron basis.

The encoder and decoder parts are defined as follows:

$$\mathbf{f}(\mathbf{x}) := ReLU(W^{enc}\mathbf{x} + \mathbf{b}^{enc}),$$

$$\hat{\mathbf{x}} := W^{dec}\mathbf{f}(\mathbf{x}) + \mathbf{b}^{dec},$$

where $\mathbf{f}(\mathbf{x}) \in \mathbb{R}^m$ is the sparse latent vector and $\hat{\mathbf{x}} \in \mathbb{R}^n$ is the reconstructed output, $W^{enc} \in \mathbb{R}^{m \times n}$, $W^{dec} \in \mathbb{R}^{n \times m}$ are the encoder and decoder matrices, respectively, $\mathbf{b}^{enc} \in \mathbb{R}^m$, $\mathbf{b}^{dec} \in \mathbb{R}^n$ are the encoder and decoder biases, respectively.

The loss is a weighted sum of the reconstruction $L^2$ loss and the $L^1$ penalty to enforce sparsity. That is

$$\mathcal{L} = \mathcal{L}_R + \lambda \mathcal{L}_P,$$

where

$$\mathcal{L}_R = \|\hat{\mathbf{x}} - \mathbf{x}\|^2$$

and

$$\mathcal{L}_P = \|\mathbf{f}(\mathbf{x})\|_1.$$

The scalar $\lambda$ is a tunable hyperparameter that controls the sparsity of the autoencoder. An important property of these equations is their homogeneity: multiplying the encoder and dividing the decoder

by any scalar results in the same reconstruction. Without additional constraints, training could *cheat* by making the sparsity loss $\mathcal{L}_P$ arbitrary small. This seemingly straightforward issue requires nontrivial solutions. Two methods have been proposed by researchers:

**Normalization** of the columns of the decoder matrix $W^{dec}$ (Bricken et al., 2023; Huben et al., 2023). This approach requires careful synchronization with gradient updates. In (Bricken et al., 2023; Gao et al., 2025), authors project away gradient information parallel to the decoder vectors to account for interactions between the Adam optimizer and normalization. In the code (Samuel Marks & Mueller, 2024), authors implement a constrained version of the Adam optimizer instead.

**Reformulation** was used instead in (Conerly et al., 2024). More precisely, they change the $L^1$ norm to the weighted norm with the weights being equal to the $L^2$ norms of the decoder columns:

$$\mathcal{L}_P = \sum_i \mathbf{f}_i(\mathbf{x}) || W^{dec}_{\cdot,i} ||_2.$$

This reformulation solves the homogeneity problem. Indeed, the trick with multiplication and division by the same scalar does not affect the proposed $\mathcal{L}_P$. In line with this best practice, we also use the reformulation method and do not normalize the decoder columns.

We have detailed this homogeneity problem because a deep understanding of underlying symmetries is crucial for identifying better inductive biases. Indeed, in the main section of our work (5), we will apply a novel reformulation method to eliminate sparse autoencoder feature bimodality.

In this paper, we primarily analyze and improve standard ReLU sparse autoencoders. However, we will also mention alternative architectures, such as TopK (Gao et al., 2025), Gated (Rajamanoharan et al., 2024), and JumpReLU (Rajamanoharan et al., 2025), which employ different activation functions and training procedures.

## 3 MEASURING ALIGNMENT BETWEEN ENCODER AND DECODER

### 3.1 ALIGNMENT SCORE - HEURISTIC MOTIVATION

According to the **linear representation hypothesis** (and the related **superposition hypothesis**), deep learning models represent concepts as directions in the activation space. SAEs learn these directions in an unsupervised manner. Indeed, each feature $i \in \{1, \ldots m\}$ corresponds to the row vector $W^{enc}_{i,\cdot}$ and the column vector $W^{dec}_{\cdot,i}$. It is not clear which of them is the *true* feature vector. In practice, the encoder vector is used for the concept detection and the decoder vector is used for the model steering (see details in (Wu et al., 2025)). Observe that intuitively these two vectors (the encoder and decoder ones) should be similar. Indeed, in (Huben et al., 2023), the authors use this rationale to tie the encoder and decoder weights, that is: (see the footnote 2 there) $W^{enc} = (W^{dec})^T$. In the tied weights scenario, both vectors coincide (assuming normalization). However, this is not a common practice, and most modern SAEs do not adopt this procedure.

Based on this intuition, we propose the following measure as a proxy for feature quality:

$$a_i = W^{enc}_{i,\cdot} \cdot W^{dec}_{\cdot,i}, \tag{1}$$

where $i \in \{1, \ldots m\}$ and $\cdot$ is the inner product in $\mathbb{R}^n$. Equivalently, these is $i$-th diagonal element of the matrix product $W^{enc}W^{dec}$. We suspect that values of $a_i$ which are negative or close to $0$ correspond to not interpretable features which are the artifact of SAE training. We refer to these scalars as **alignment scores**.

While the motivation for this section is based on heuristics, we also provide more precise arguments for introducing this score, grounded in an analysis of toy models, in the next subsection. One of the main reasons for using the inner product formulation is the homogeneity property described in the previous subsection. Note that the alignment score (Equation 1) remains invariant with respect to multiplication of the encoder and division of the decoder by the same scalar. We emphasize that preserving this structural algebraic property is crucial in constructing the alignment score. This motivates the choice of the inner product over, perhaps more intuitive, cosine similarity, which exhibits stronger scale invariance with respect to both variables.

## 3.2 Alignment Score Should be Close to 1 - Toy Model Argument

In this subsection we will show that the alignment score is not just an ad-hoc formula and show that it should be cluster close to 1. This will be confirmed empirically in the experiments.

Let us consider one of the simplest toy model: the activation space is two dimensional, that is $n = 2$ and there is only one feature in a sparse autoencoder $m = 1$. Moreover, we ignore biases and use the training set of just a one single point $\mathbf{x} \in \mathbb{R}^2$. We also ignore the sparsity penalty, so the setting is:

$$f(\mathbf{x}) = ReLU(w^{enc} \cdot \mathbf{x}),$$

$$\hat{\mathbf{x}} = f(\mathbf{x})w^{dec},$$

where $w^{enc}$ and $w^{dec}$ are vectors and $\cdot$ denotes an inner product in $\mathbb{R}^2$. The latent $f(\mathbf{x})$ is just a scalar because we set $m = 1$.

Naturally, this is a very artificial setting and the autoencoder can just learn the identity. However, as we will see, the ReLU nonlinearity makes the training dynamics not that trivial. Now let us check, when this toy model can perfectly reconstruct the input, that is when $\hat{\mathbf{x}} = \mathbf{x}$. We have:

$$\mathbf{x} = ReLU(w^{enc} \cdot \mathbf{x})w^{dec}$$

Observe that $\mathbf{x}$ and $w^{dec}$ must be parallel, so $\mathbf{x} = \alpha w^{dec}$ for some scalar constant $\alpha$. Hence we obtain

$$\alpha w^{dec} = ReLU(\alpha w^{enc} \cdot w^{dec})w^{dec}.$$

This leads to our desired equation:

$$1 = w^{enc} \cdot w^{dec}.$$

In fact, this single training point example was the first "experiment" done in this research project and the connection with the MCS and the bimodality phenomenon were discovered later.

The presented toy experiment is very crude and simplified, but as we will see, it scales effectively to real sparse autoencoders. It mirrors the illustrative example from (Benjamin Wright, 2024), which the authors used to explain the feature suppression/shrinkage effect.

## 4 Preliminary Experiments

### 4.1 Experiment 1: Alignment Scores Are Bimodal

In the first experiment, we calculated the histograms of the alignment scores. The results are presented in Figure 1. Across different models and SAE architectures, we observed that the score distribution is consistently bimodal. We suspect that this bimodality is similar to the phenomenon discovered by (Huben, 2023). Additionally, a footnote in the Anthropic work (Bricken et al., 2023) speculates that this same bimodality corresponds to feature frequency. It is important to emphasize that, unlike MCS, the alignment score is calculated without training an additional larger SAE. Furthermore, unlike feature frequency, it is derived directly from the weights without requiring additional language corpus data.

### 4.2 Experiment 2: Alignment Scores Are Highly Correlated with MCS

In the next experiment, we compared the MCS scores with the alignment scores for an autoencoder trained on MLP activations from the second layer of Pythia (Biderman et al., 2023). We utilized the SAE from (Riggs, 2023), where SAEs of different sizes are available. This approach avoided the need to train a larger autoencoder from scratch, which would otherwise be required to calculate MCS. The resulting scatter plot is shown in Figure 2. The Pearson correlation coefficient is high, with a value of $0.65$. Additionally, as observed, low alignment scores correspond to weaker features that were not discovered by a larger interpreter model. The cluster of the best features concentrates at an alignment value of $1.0$ as predicted by the toy model argument from 3.2.

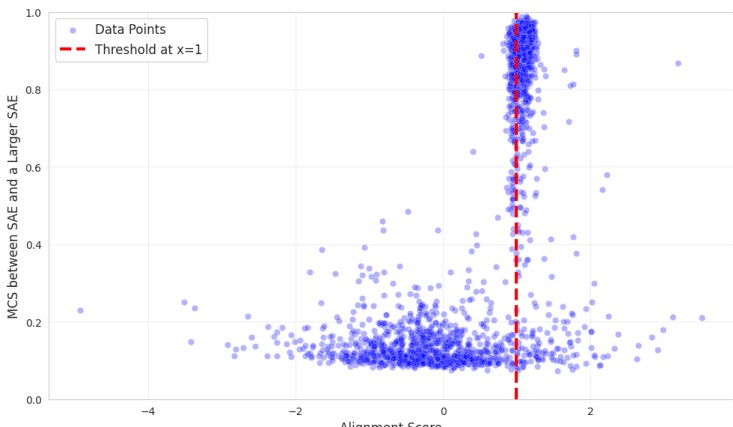

Figure 2: The scatter plot of MCS vs the alignment score. The Pearson correlation is high and equal to 0.65. The red vertical line denotes the alignment score value equal to 1.

### 4.3 EXPERIMENT 3: ALIGNMENT SCORES CAN BE USED TO FIND INTERPRETABLE FEATURES

In the previous experiment, we compared one proxy with another. Here, we directly assess whether alignment scores can be used to evaluate feature interpretability. We apply the commonly used autointerpretability protocol introduced in (Bills et al., 2023). In this method, LLM judge first generates a natural language description of a given neuron or direction based on a sample of maximal activations. Then, the judge model uses this generated description to score each token in another sample of texts. The interpretability score is the Pearson correlation between the LLM judge scores per token and the actual activation scores. In the case of SAEs, the activation score for feature $i$ is equal to the encoder value $\mathbf{f}_i(\mathbf{x})$. For more details on autointerpretability protocols, see (Paulo et al., 2025).

The resulting scatter plot comparing the alignment scores with autointerpretability is presented in figure 3. The dead features for which we did not have enough non-zero activation to run the autointerpretability are presented as black dots.

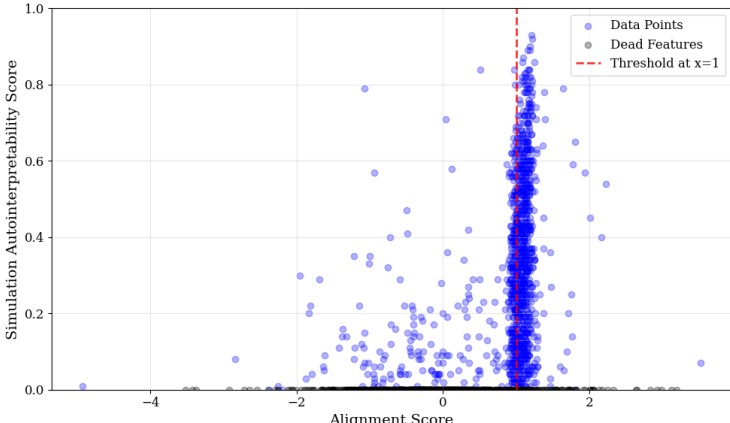

Figure 3: Autointerpretability scores vs the alignment score. The red vertical line denotes the alignment score value equal to 1. The Pearson correlation is equal to 0.32.

## 5 Fixing Bimodality

From a practical standpoint, having a measure of feature quality can aid in designing more effective training algorithms for SAEs. This idea has been suggested in (Cunningham & Riggs, 2023) in the context of MCS: *"Also, the bimodal nature of the MCS scores suggests that we might be able to use unorthodox training strategies where we identify the convergent features and then perhaps freeze them while aggressively perturbing the remaining vectors"* While implementing this with the demanding MCS score was challenging, the overall simplicity of the proposed alignment score makes it feasible. Additionally, feature-aligned sparse autoencoders (Marks et al., 2024) employ a similar approach, requiring the training of two sparse autoencoders in parallel and resampling. In contrast, we propose a straightforward algebraic transformation. As discussed in 2, the encoder and decoder are defined as follows:

$$\mathbf{f}(\mathbf{x}) := ReLU(W^{enc}\mathbf{x} + \mathbf{b}^{enc}),$$
$$\hat{\mathbf{x}} := W^{dec}\mathbf{f}(\mathbf{x}) + \mathbf{b}^{dec},$$

Now, the key idea is to leave $\mathbf{b}^{enc}, \mathbf{b}^{dec}$ and $W^{dec}$ as arbitrary trainable parameters and modify $W^{enc}$. In the proposed training method we instead of $W^{enc}$ use a trainable parameter matrix $A \in \mathbb{R}^{m \times n}$ and define an encoder row $W^{enc}_{i,\cdot}$ as:

$$W^{enc}_{i,\cdot} := A_{i,\cdot} + \alpha_i W^{dec}_{\cdot,i}, \tag{2}$$

where

$$\alpha_i = \frac{1 - A_{i,\cdot} \cdot W^{dec}_{\cdot,i}}{||W^{dec}_{\cdot,i}||^2}.$$

It is straightforward to check that:

$$W^{enc}_{i,\cdot} \cdot W^{dec}_{\cdot,i} = 1, \tag{3}$$

for every $i = 1, \ldots, m$.

Now, we just use the standard gradient descent training for parameters $b^{enc}, b^{dec}, W^{dec}$ and $A$.

In essence, we constrain the manifold of possible weights by enforcing the equation 3. Our conjecture is that this inductive bias can enhance the training of the sparse autoencoder. This approach is similar in spirit to convolutional networks. Indeed, every convolutional layer is essentially a linear layer constrained by the local receptive field (Fukushima, 1988) and weight sharing (Lecun et al., 1998). In theory, a linear layer could learn convolution during training, but incorporating this inductive bias makes the training of computer vision models more effective.

We will refer to the proposed method as **aligned training** and demonstrate that it outperforms standard ReLU autoencoders on several benchmarks across different sparsities, dictionary sizes, and underlying language models.

Moreover, because equation 3 constrains every encoder row to lie on an affine hyperplane of dimension $n-1$, we can set the last column of $A$ to 0 (and not train it). The resulting model can still express the same space of weights. In other words, we achieve **compression for free**; an autoencoder from aligned training has slightly fewer parameters.

### 5.1 Experiment 4: Aligned Training Achieves Pareto Improvement on Reconstruction Metrics

The most basic metric used to evaluate the quality of any autoencoder is its ability to reconstruct the input. For comparing sparse autoencoders, it is essential to measure reconstruction across different sparsities, as less sparse autoencoders can trivially achieve better reconstruction (and, in the limit, even learn the identity function). The sparsity is measured by $L_0$ "norm", which is the number of nonzero values in the encoded vector $\mathbf{f}(\mathbf{x})$.

Following (Bussmann et al., 2024; Karvonen et al., 2025) we use two methods to check how good is the the autoencoder reconstruction: the explained variance and the recovered cross entropy loss (consult the Appendix A for precise formulas of these metrics).

We trained the ReLU autoencoders on the 50 million tokens from the Pile (Gao et al., 2020) using both the standard and the aligned methods and performed the test on OpenWebText. The results are presented in the figure 4. The aligned method outperforms standard training across two models, four sparsities and three dictionary sizes (see figures 16, 18 in the Appendix D.5 for 16K and 65K dictionary sizes).

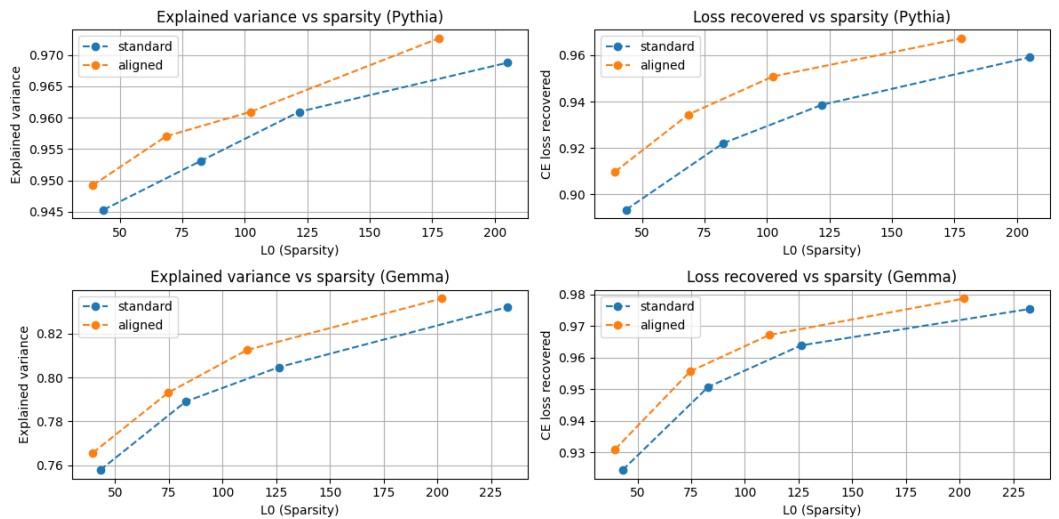

Figure 4: The proposed training methods improves the recovered cross entropy of sparse autoencoder across different sparsities. The results for the dictionary size $4096$ for the 8th layer of the Pythia 160M and the 12th layer of Gemma 2 2B (Team et al., 2024)

## 5.2 EXPERIMENT 5: ALIGNED TRAINING AS A PARAMETER-FREE METHOD TO AVOID DEAD FEATURES

One of the notorious issues in training sparse autoencoders is the problem of dead features, where a feature does not activate for any token (Bricken et al., 2023; Gao et al., 2025; Jermyn & Templeton, 2024). This occurs because it is an easy way to minimize the sparsity loss. Several advanced measures have been developed to reduce the number of dead features:

- Resampling schemes with non-uniform probabilities (Bricken et al., 2023).
- Adding artificial gradients, known as *ghost grads*, which were introduced briefly by Anthropic (Jermyn & Templeton, 2024) but later abandoned due to their role in causing loss spikes (Adly Templeton & Henighan, 2024).
- Introducing an auxiliary loss term for dead features, as proposed in the TopK autoencoder paper (Gao et al., 2025).

We emphasize that all these methods require extensive hyperparameter tuning. In contrast, our proposed aligned training completely eliminates dead features: see Figure 5. The aligned training is parameter-free and does not require sampling or other non-differentiable operations that could destabilize training. As shown, the standard ReLU sparse autoencoder has approximately 20% dead neurons, while the aligned autoencoder has none. Furthermore, resampling and ghost grad methods focus on resurrecting dead features, whereas aligned training constrains the parameter space to prevent features from dying in the first place.

## 5.3 EXPERIMENT 6: SCALING TRAINING TO 500M TOKENS AND COMPARING TO STATE-OF-THE-ART

In this section, we compare aligned training to state-of-the-art ReLU sparse autoencoders provided by SAEBench. To ensure a fair comparison, we scale our training to **half a billion tokens** from The

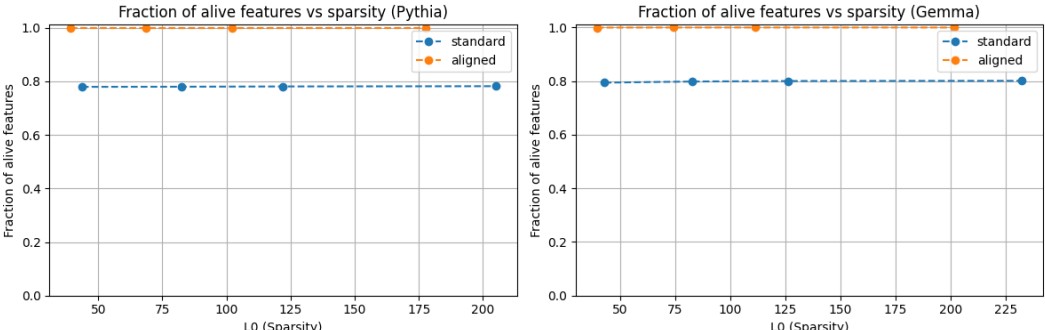

Figure 5: The proposed training method is a parameter-free approach significantly reducing the number of dead features. The results for the dictionary size 4096 for the 8th layer of the Pythia 160M and 12th layer of Gemma 2 2B.

Pile (Gao et al., 2020), matching the compute used in SAEBench (recall that our formula was derived from a **1-token** toy model). As shown in Figure 6, our training surpasses both the standard SAEs trained by us and those provided by the benchmark authors. This demonstrates that the advantage is robust and not a result of cherry-picking hyperparameters. Additionally, Figure 7 shows that our training achieves a significantly lower number of dead features.

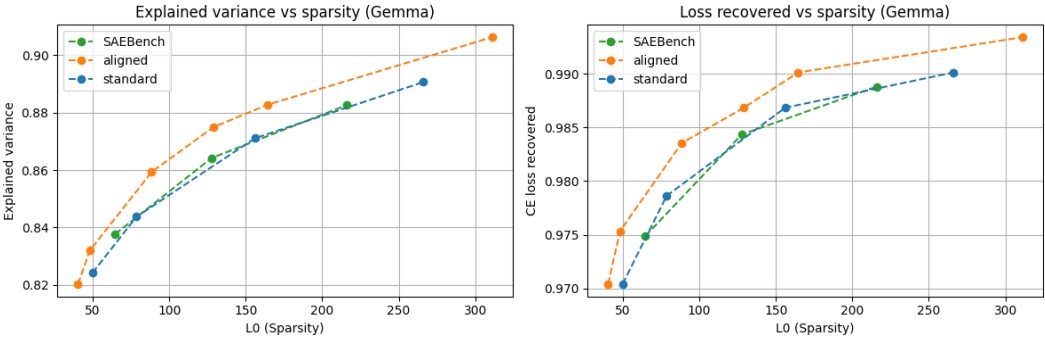

Figure 6: Reconstruction metrics comparing our method to standard (trained by us) and state-of-the-art checkpoints provided by SAEBench. Training was conducted on half a billion tokens for layer 12 of Gemma 2 2B with a dictionary size of 65K features.

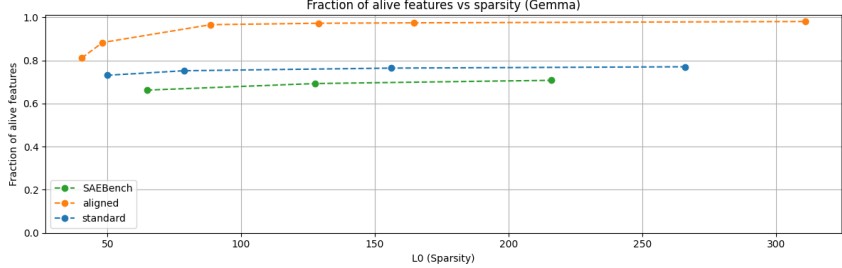

Figure 7: Fraction of alive features comparing our method to standard (trained by us) and state-of-the-art checkpoints provided by SAEBench. Training was conducted on half a billion tokens for layer 12 of Gemma 2 2B with a dictionary size of 65K features.

## 5.4 EXPERIMENT 7: ALIGNED TRAINING IMPROVES TOPK AND BATCHTOPK AUTOENCODERS

So far, we focused on standard sparse autoencoders (sometimes referred to as *vanilla* (Rajamanoharan et al., 2025)). As shown in the first figure, bimodality also appears for different activation functions. In this section we provide the results for applying the aligned training method for the newer architecture of TopK sparse autoencoder (Gao et al., 2025) and its variant BatchTopK (Bussmann et al., 2024). In this approach, the sparsity is enforced not by the $L^1$ penalty, but by the TopK activation applied on top of $\mathbf{f}(\mathbf{x})$ (as we described in the section 2). Notice, that our training reformulation 5 can be directly applied also in this case. While the primary motivation and toy model argument were based on the ReLU model, it turns out that similar improvements can be achieved even with the TopK approach.

Figure 8 presents the results of training TopK autoencoders on 50 million tokens from The Pile using both the standard TopK and the aligned method. The proposed training method outperforms the baseline in the low-sparsity regime, while for higher sparsities, there is no significant difference between the two. Furthermore, the same pattern is observed in the number of dead neurons, as shown in Figure 9.

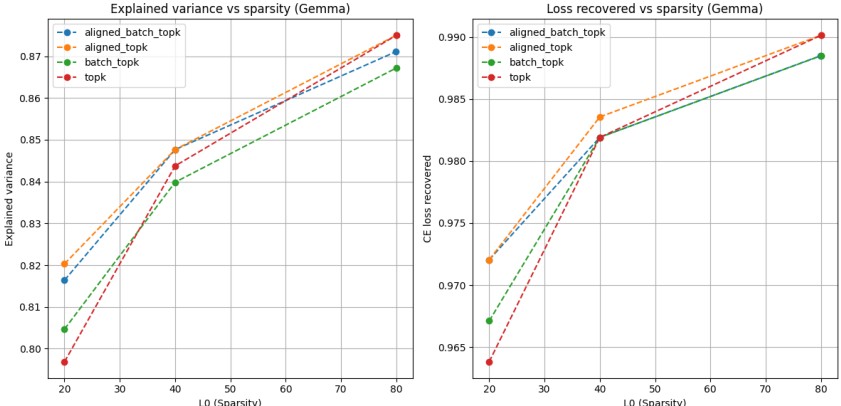

Figure 8: The proposed training methods improves the TopK and BatchTopK autoencoders. The results for the dictionary size $65K$ for the 12th layer of Gemma 2 2B.

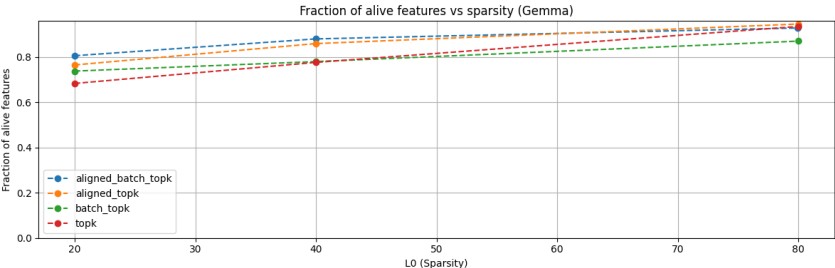

Figure 9: The proposed training method can be applied off-the-shelf the TopK and BatchTopK autoencoders, when it reduces the fraction of dead neurons. The results for the dictionary size $65K$ for the 12th layer of Gemma 2 2B

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

## A    FORMULAS FOR THE RECONSTRUCTION METRICS

Explained variance is defined as

$$1 - \frac{\frac{1}{K}\sum_{k=1}^{K} ||\mathbf{x}_k - \hat{\mathbf{x}_k}||^2}{\frac{1}{K}\sum_{k=1}^{K} ||\mathbf{x}_k - \boldsymbol{\mu}||^2},$$

where $\boldsymbol{\mu}$ is the mean activation vector:

$$\boldsymbol{\mu} = \sum_{k=1}^{K} \mathbf{x}_k.$$

Cross entropy loss recovered is equal to

$$\frac{H^* - H_0}{H_{orig} - H_0},$$

where $H_{orig}$ refers to the cross-entropy loss calculated for the model's next-token prediction task. $H^*$ represents the cross-entropy loss obtained by replacing the model's activation $\mathbf{x}$ with its SAE-reconstructed version $\hat{x}$ during the forward pass. Additionally, $H_0$ denotes the cross-entropy loss that results from zeroing out the activation $\mathbf{x}$.

## B    TECHNICAL DETAILS

In this appendix, we provide all the technical details necessary to replicate the conducted experiments. For loading pretrained SAEs, we used the sae-lens framework (Bloom et al., 2024). In histograms 1, we utilized the following SAEs:

- For Pythia: Model 70M, residual stream at layer 3. Sae-lens release: *pythia-70m-deduped-res-sm*, id: *blocks.3.hook_resid_post*.
- For LLaMA: Model 3 8B IT, residual stream at layer 25. Sae-lens release: *llama-3-8b-it-res-jh*, id: *blocks.25.hook_resid_post*.
- For Gemma: Model 2 2B, residual stream at layer 12. Sae-lens release: *sae_bench_gemma-2-2b_topk_width-2pow12_date-1109*, id: *blocks.12.hook_resid_post_trainer_0*.

For scatter plots 2 and 3 we used the ReLU autoencoders from (Riggs, 2023): *https://huggingface.co/Elriggs/autoencoder_layer_2_pythia70M_5_epochs*.

For autointerpretability we used the code *https://github.com/HoagyC/sparse_coding* which we modified to use the open source model Gemma 3 27B IT as a judge instead of the close source GPT.

For the remaining experiments, we used the same settings as in SAEBench, specifically layer 8 of Pythia 160M and layer 12 of Gemma 2 2B.

## C CODE IMPLEMENTATION

For training our autoencoders, we utilized the code (Samuel Marks & Mueller, 2024) and implemented the key transform 2 using the Python function described below. For the calculating the inner products per feature, we employed the einops (Rogozhnikov, 2022) library, which provides flexible Einstein notation operations on tensors.

```python
def get_the_encoder_matrix(dict_size: int,
↪  encoder_weights_orthogonal_part: Float[Tensor,
↪  "activation_dim-1 dict_size"], decoder_weights: Float[Tensor,
↪  "dict_size activation_dim"]) -> Float[Tensor, "activation_dim
↪  dict_size"]:
    zeros = torch.zeros(1,
    ↪  dict_size).to(encoder_weights_orthogonal_part)
    appended = torch.concat([encoder_weights_orthogonal_part,
    ↪  zeros])
    inner_products = einops.einsum(decoder_weights, appended,
    ↪  "dict_size activation_dim, activation_dim dict_size ->
    ↪  dict_size")
    decoder_norms_squared = decoder_weights.pow(2).sum(dim=1)
    reparametrized = appended + decoder_weights.T * (1 -
    ↪  inner_products) / decoder_norms_squared
    return reparametrized
```

## D ADDITIONAL EXPERIMENTS

### D.1 ALIGNMENT TRAINING IS NOT JUST WEIGHT-TYING

As previously noted, a stronger condition can be employed to align the encoder and decoder vectors. The weight tying approach (Huben et al., 2023) assumes that the encoder and decoder vectors are identical. To evaluate these methods, we conducted experiments comparing the standard approach, weight tying, and aligned training. The results presented on 10 and 11 indicate that:

- Weight tying significantly reduces the number of dead neurons but compromises reconstruction quality.

- Aligned training enhances both reconstruction performance and the proportion of active features.

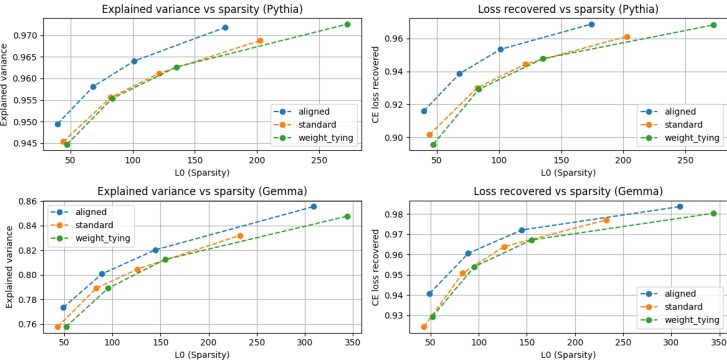

Figure 10: Reconstruction metrics for Pythia 160M layer 8 and Gemma 2 2B layer 12 and the dictionary size of 16384 features. Weight tying underperforms.

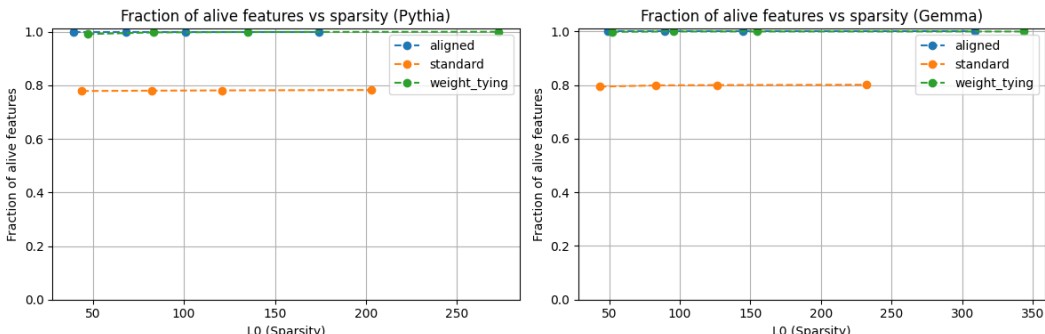

Figure 11: For Pythia 160M (layer 8) and Gemma 2B (layer 12) with a dictionary size of 16,384 features, weight tying also effectively mitigates dead features but results in suboptimal reconstruction quality.

## D.2    THE ALIGNED TRAINING CAN BE COMBINED WITH P-ANNEALING

There are several non-standard methods for training ReLU autoencoders, such as square root (Logan Riggs Smith, 2024), tanh (Jermyn et al., 2024), and p-annealing (Karvonen et al., 2024b). These methods were introduced to address the issue of feature shrinkage (Benjamin Wright, 2024). Our work was motivated by a different issue—bimodality and low-quality/dead features. It is instructive to compare these approaches and explore whether they overlap or achieve synergy when applied together. As presented in the figure 12 and 13, there is indeed the improvement.

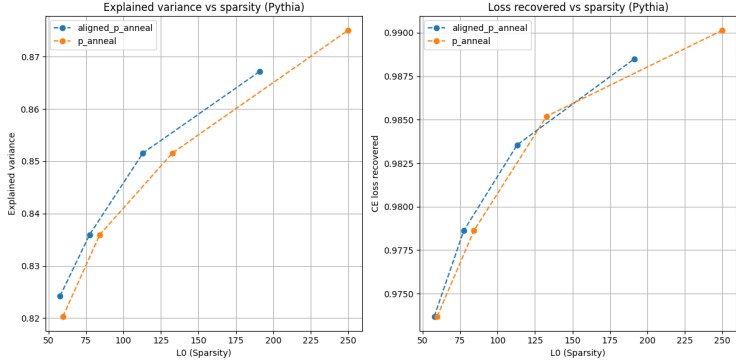

Figure 12: Reconstruction metrics for Pythia 160M layer 8 and the dictionary size of 4096 for SAE runs with 3 random seeds.

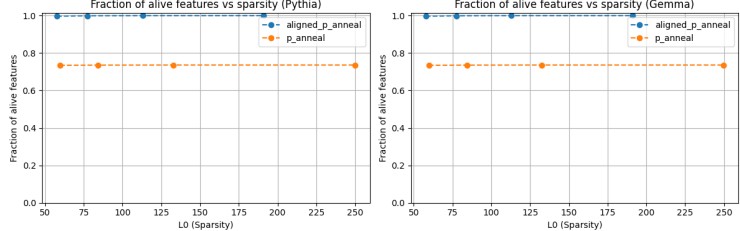

Figure 13: Fraction of alive neurons for Pythia 160M layer 8 and Gemma 2 2B layer 12 and the dictionary size of 16384 features.

### D.3 THE RESULTS ARE ROBUST WITH RESPECT TO DIFFERENT SEEDS

In this section we perform the important validation. There is an inherent stochasticity in training sparse autoencoders, hence it is crucial to check how the results change with varying random seeds. As demonstrated on the figure 14 the difference is negligible.

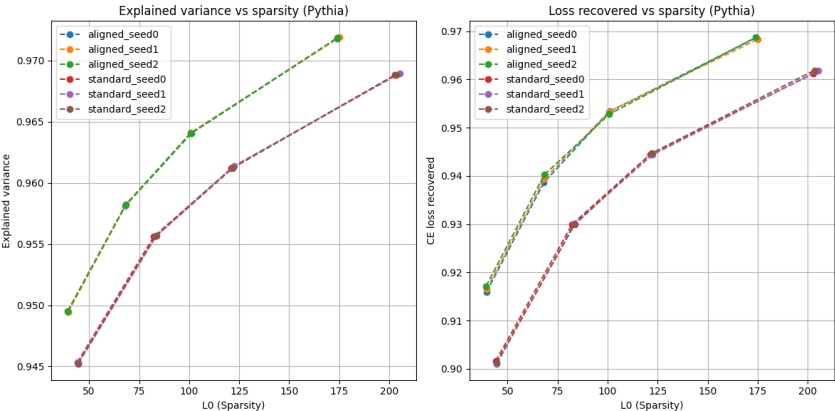

Figure 14: Reconstruction metrics for Pythia 160M layer 8 and the dictionary size of 4096 for SAE runs with 3 random seeds.

### D.4 ALIGNED TRAINING OUTPERFORMS STANDARD AUTOENCODER IN SPURIOUS CORRELATION REMOVAL

The metrics used in the two previous experiments are still theoretical proxies for evaluating the usefulness of an autoencoder. We also tested our proposed method on a practical downstream task and found that it achieves better results. The application of sparse autoencoders to remove spurious correlations was introduced in the SHIFT method (Marks et al., 2025) in the context of gender bias (De-Arteaga et al., 2019) and extended in (Karvonen et al., 2024a). We used the implementation of this metric from the SAEBench suite (Karvonen et al., 2025).

As shown in Figure 15, aligned training outperforms the standard method on the Pythia model. For Gemma, the two methods are comparable for smaller sparsity ranges, but our method shows a clear advantage for larger $L_0$ values.

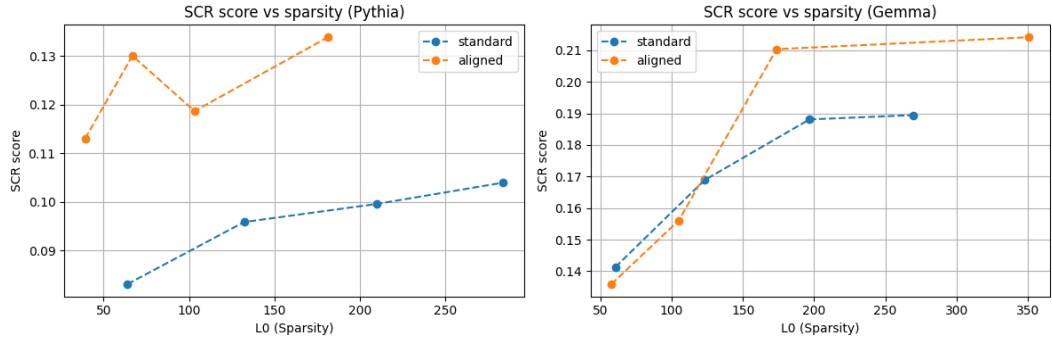

Figure 15: Results for spurious correlation removal metric from SAEBench on the dictionary size = 65K for Pythia 160M and Gemma 2 2B.

D.5 RESULTS ARE CONSISTENT FOR DIFFERENT DICTIONARY SIZES

In this subsection we show the reconstruction loss and dead neurons metrics for the different dictionary sizes. For 16K features consult the Figures 16 and 17. We also conducted the experiments for the large dictionary of 65K, see the Figures 18 and 19.

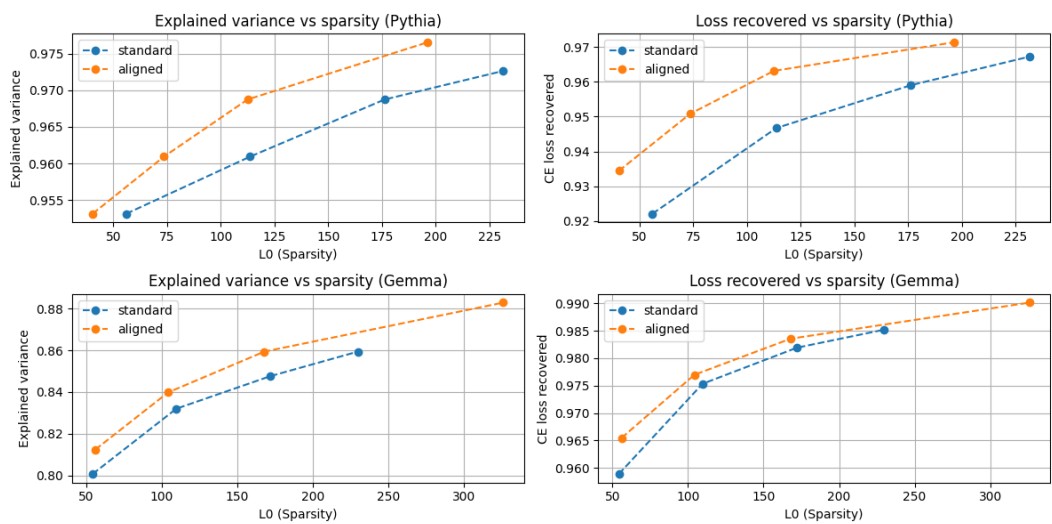

Figure 16: Reconstruction metrics for Pythia 160M layer 8 and Gemma 2 2B layer 12 and the dictionary size of 16384 features.

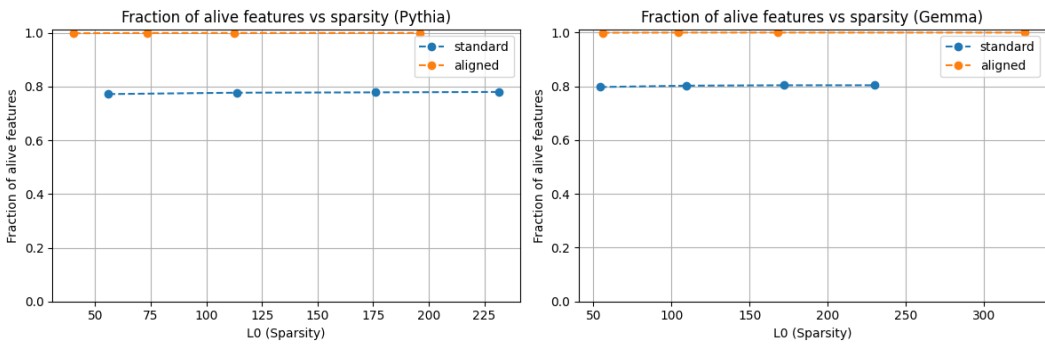

Figure 17: Fraction of alive neurons for Pythia 160M layer 8 and Gemma 2 2B layer 12 and the dictionary size of 16384 features.

## E CONNECTION WITH THE FEATURE CONSISTENCY

There is growing interest in measuring the feature consistency of sparse autoencoders, as seen in recent works (Paulo & Belrose, 2025; Song et al., 2025). This issue is closely related to the MCS metric we discussed, as it evaluates the similarity between SAEs with the same dictionary sizes trained on different seeds.

Archetypical SAEs Fel et al. (2025) address this phenomenon by training models to fit the convex hull of the data points, improving feature consistency. However, this approach comes at the cost of degrading reconstruction metrics. This trade-off highlights a key distinction between their solution and ours: aligned training improves reconstruction metrics and also reduces the fraction of dead neurons.

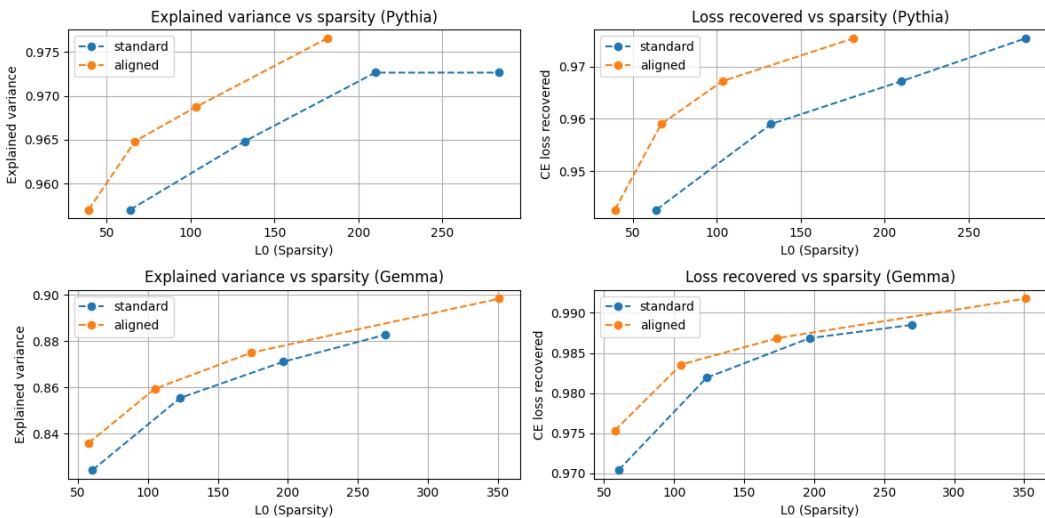

Figure 18: Reconstruction metrics for Pythia 160M layer 8 and Gemma 2 2B layer 12 and the dictionary size of 65K features.

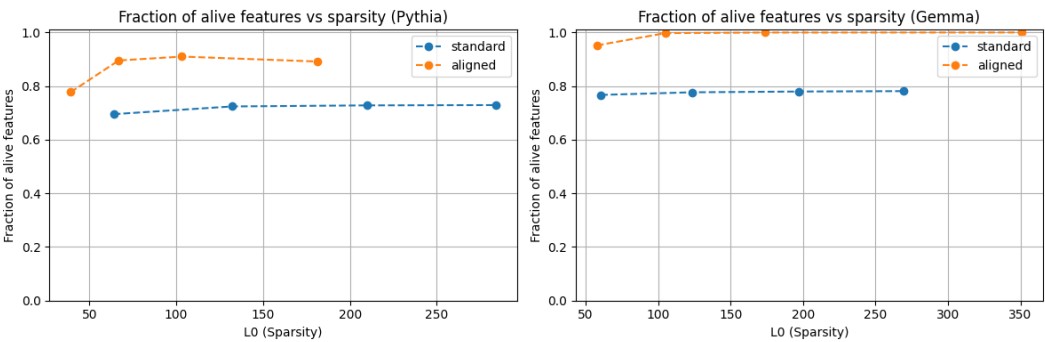

Figure 19: Fraction of alive neurons for Pythia 160M layer 8 and Gemma 2 2B layer 12 and the dictionary size of 65K features.

In follow-up experiments, we plan to evaluate how our method performs against archetypical SAEs on their proposed stability metric.

## F    THE BIMODALITY DYNAMICS

In this experiment, we loaded the SAEBench checkpoints from different training steps and produced the histograms of the alignment score across time, see Figure 20.

## G    SANITY CHECK

We also have performed a trivial sanity check that for the alignment score is indeed 1 for our training method, see the Figure 21.

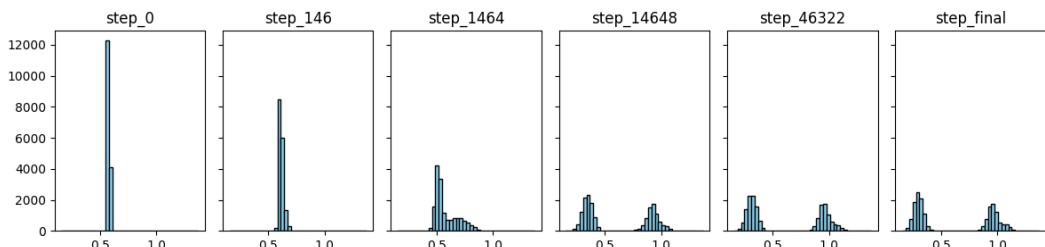

Figure 20: Creation of the bimodality during training of a TopK SAE trained on the layer 12 of Gemma 2 2B. We used checkpoints provided in SAEBench(Karvonen et al., 2025). We hypotesize that this is caused by two competing forces: useful features are pushed to the cluster with the alignment close to 1. On the other hand, useless features are pushed to the left, where they rarely activate. The second effect is caused by the sparsity goal.

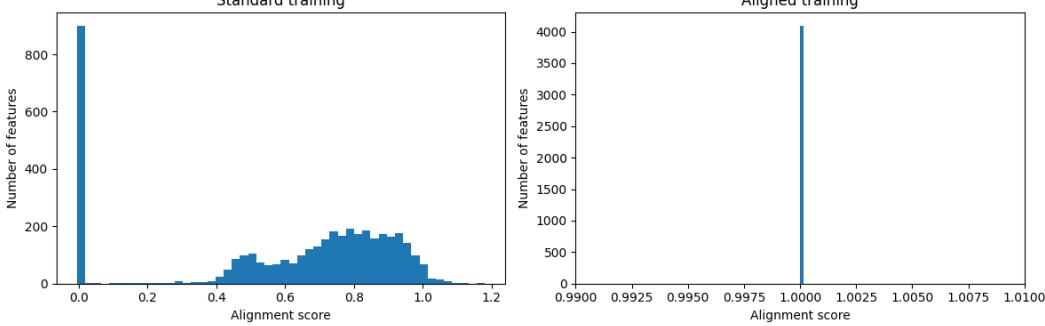

Figure 21: Histograms of alignment scores of the standard and aligned trained sparse autoencoders. The plot on the right serves as a sanity check: the proposed reformulation achieves its intended goal by forcing all alignment scores to be equal to 1, effectively preventing the creation of bimodality.

