# OpenReview forum: "Bimodality of Sparse Autoencoder Features is Still There and Can Be Fixed"
_ICLR.cc/2026/Conference — Submitted to ICLR 2026_

### Official Review · Reviewer_72nK · 2025-10-21

**Soundness:** 3
**Presentation:** 2
**Contribution:** 2
**Rating:** 4
**Confidence:** 4

**Summary:**

The authors resurface a phenomena described in some of the first works applying sparse autoencoders (SAEs) to large language models. The maximum cosine similarity (MCS) of SAE features between two dictionaries tend to have a bimodal distribution, with a peak close to 1 and a second peak close to what would be expected from a random distribution.
The authors propose a simpler and cheaper proxy than MCS and show that current SAEs exibit bimodality in this proxy metric. They show that the proxy is correlated with MCS and alignment scores, and propose a new method for training SAEs that decrease this bimodality.

**Strengths:**

The bimodality of MCS in SAEs is still an interesting subject to explore and improve on.
A proxy for SAE quality that does not require any data or training an extra SAE is interesting.
The proposed training technique improves over regular L1 training.

**Weaknesses:**

While it is true that people have stopped looking at the bimodality of MCS between SAEs of different sizes, there have been other work describing this problem between different SAE seeds (Sparse Autoencoders Trained on the Same Data Learn Different Features, Position: Mechanistic Interpretability Should Prioritize Feature Consistency in SAEs) as well as describing methods to improve these metrics (Archetypal SAE: Adaptive and Stable Dictionary Learning for Concept Extraction in Large Vision Models) and none of this work is mentioned.

The quality of all figures needs to be improved. The main track of ICLR deserves more than default matplotlib plots.

While the authors report a high correlation between MCS and their alignment metric their proxy does not seem to be very predictive of MCS.

Figure 3 is impossible to interpret as the authors do. 'As expected, the scores close to one correspond to the more interpretable
features'. The spread of scores of features with aligment close to one covers all the range shown.

Improving over 'standard' SAEs is no longer satisfactory. There are several architectures that beat standard SAEs. 50M tokens is also a very short training run.

**Questions:**

-Why use the inner product instead of cosine similarity?
-Where do the models from figure 1 come from?

---

> ### Author Response · Authors · 2025-11-19
>
> We are thankful for your time and help, especially related to the related works. We were glad to hear that you emphasized that our score doesnot require any training data.
>
> ## 1. Our method improves modern SAEs too
>
> > Improving over 'standard' SAEs is no longer satisfactory. There are several architectures that beat standard SAEs.
>
> We acknowledge this point and have conducted a new experiment on state-of-the-art TopK and BatchTopK sparse autoencoder architectures. Our approach not only improves reconstruction loss but also significantly reduces the number of dead neurons. These findings are presented in subsection 5.4 of the revised paper, further strengthening our claims.
>
> ## 2. Short run
>
> > 50M tokens is also a very short training run.
>
> The 50M token budget was chosen based on SAEBench [1], Section 4.4, which suggests that SAEs generally converge at this stage. To ensure a robust comparison, we also performed a full 500M token training run, aligning our results with those from SAEBench. These details are provided in Section 5.3 of the revised manuscript.
>
>  ## 3. Why not cosine similarity
>
> > -Why use the inner product instead of cosine similarity
>
> Interestingly, the equation cossim() = 1 leads to the weight tying training when the encoder and decoder vector are identical. While this approach also reduces the number of the dead neurons, it harms the autoencoder reconstruction. We performed this comparison as a new experiment added in the section E1 of the appendix
>
> ## 4. SAE from figure 1
>
> > -Where do the models from figure 1 come from?
>
> The technical details about the used SAEs are in the Appendix. In the revised version, it is section B. We have loaded the available SAE checkpoints from sae-lens, so it can be replicated.
>
> ## 5. Why we only demonstrated the correlation with MCS
>
> >While the authors report a high correlation between MCS and their alignment metric their proxy does not seem to be very predictive of MCS.
>
> We deliberately do not focus heavily on the MCS score, as it was introduced by Sharkey et al. [2] as a simplistic approximation of feature quality when ground truth is unavailable. Although the correlation between the alignment score and MCS provides a compelling rationale for using the alignment score in training, our aim was never to fully account for MCS.
>
>  ## 6. Correlation with autointerpretability
>
> > Figure 3 is impossible to interpret as the authors do. 'As expected, the scores close to one correspond to the more interpretable features'. The spread of scores of features with aligment close to one covers all the range shown.
>
> Thanks, we changed the description to just emphasize the nontrivial correlation in the revised version. The autointerpretability metric is a noisy LLM-as-a-judge method and we do not expect a cleaner signal than a correlation.
>
> ## 7. Quality of figures
>
> > The quality of all figures needs to be improved. The main track of ICLR deserves more than default matplotlib plots.
>
> Thanks, we prettified the histograms and scatter plots with seaborn.

---

> ### Author Response · Authors · 2025-11-19
>
> ## 8. Related works
>
> > While it is true that people have stopped looking at the bimodality of MCS between SAEs of different sizes, there have been other work describing this problem between different SAE seeds (Sparse Autoencoders Trained on the Same Data Learn Different Features, Position: Mechanistic Interpretability Should Prioritize Feature Consistency in SAEs) as well as describing methods to improve these metrics (Archetypal SAE: Adaptive and Stable Dictionary Learning for Concept Extraction in Large Vision Models) and none of this work is mentioned.
>
>
>
> Thank you for the references. We have added an extended related work section in Appendix E to address these points.
>
> The recent surge of interest in feature consistency (2025) indeed adds nuance to the discussion. However, we still argue that the bimodality phenomenon, first highlighted in the MCS-related works of 2023, was overlooked in the seminal SAE papers from late 2023 and 2024. For instance, in the work you mentioned [3], the authors use the Hungarian algorithm to align SAE features: a technique standard in 2023 MCS works [4]. This underscores our point that this line of ideas has been neglected.
>
> Regarding Archetypical SAEs, they cannot serve as an alternative to our method, as they improve feature consistency at the expense of reconstruction metrics, whereas our approach enhances both reconstruction metrics and reduces dead neurons. Naturally, this raises the reverse question: can our simpler aligned training method supersede Archetypical SAEs? This is an intriguing conjecture we are actively investigating. Calculating their proposed stability score for our method is on our to-do list.
>
>
>
>
>
> We appreciate your thoughtful feedback and the new references, which have been invaluable in helping us contextualize our work within the broader field. Your insights have significantly enriched our understanding and strengthened our manuscript. Thank you for your contributions!
>
> [1] SAEBench: A Comprehensive Benchmark for Sparse Autoencoders in Language Model Interpretability
>
> [2] Taking features out of superposition with sparse autoencoders
>
> [3] Sparse Autoencoders Trained on the Same Data Learn Different Features
>
> [4] https://github.com/HoagyC/sparse_coding/blob/main/interp_notebooks/dict_compare.ipynb

---

> ### Comment · Reviewer_72nK · 2025-11-27
>
> I think a large question remains, for me, unanswered with respect to the motivation of this work. If feature alignment is such a desired property why shouldn't we just train tied SAEs? If there is a conflict between the reconstruction objective and the alignment of features, why should the alignment of features be a good measure for their quality? I'm not convinced that the toy model is relevant for any practical case.
>
> > Thanks, we changed the description to just emphasize the nontrivial correlation in the revised version. The autointerpretability metric is a noisy LLM-as-a-judge method and we do not expect a cleaner signal than a correlation.
>
> I agree with the authors that autointerpretability is a noisy measure of feature quality. Given that the authors have not given any other measure of feature quality I don't think there is enough evidence that aligning features is correct or helpful.
>
> > The 50M token budget was chosen based on SAEBench [1], Section 4.4, which suggests that SAEs generally converge at this stage. To ensure a robust comparison, we also performed a full 500M token training run, aligning our results with those from SAEBench.
>
> I think this is a misrepresentation of SAEBench, specially when they explicitly say 'We caution that minor quantitative improvements may also have major qualitative improvements, and higher training budgets may be worthwhile, especially
> for larger dictionary sizes.'
>
> I'm concerned that the batch topk SAEs seem to be underperforming normal topk SAEs, any reason for this? I think sligtly  high L0s would also be important to consider, but I understand budget constraints with respect to training SAEs.

---

> > ### Author Response · Authors · 2025-11-28
> >
> > Thanks for the comment with clarification.
> > - The core value of the paper is introducing a very cheap parameter-free method of improving SAEs on both the reconstruction metrics and (significantly) in the number of alive neurons. We agree that we should that have made this point more clear. To cite another reviewer: "I started reading the paper thinking it was about the MCS score and bi-modality, to discover that actually it's about this new SAE architecture that is much more exciting and interesting. This should be stated up-front! If you have created a new SAE architecture that seems to solve real problems with dead-latents and achieves good performance, you should lead with that!".
> > - The argument that aligning features is helpful is given by using the most common method in SAE literature which is by showing the better reconstruction-sparsity tradeoff. (for example the same argument used in BatchTopK paper)
> > - As for BatchTopK vs TopK, we did not found consistent difference between them, sometimes one is better than the other. We were unable to replicate the claim from BatchTopK paper that the improvement is uniform. To check that this is not a bug on our side, we also checked plots on Neuronpedia and we do not see the consistent advantage of using BatchTopk over TopK.
> >
> > In other words, we believe that the method is of practical importance even without the "alignment" narrative, toy model and motivation. We could change the name "aligned training" to "gobblydock training" and the core claim will still be valid. Moreover, our results show that there is no "conflict between the reconstruction objective and the alignment of features", quite the contrary.

---

> ### Author Response · Authors · 2025-11-28
>
> >If feature alignment is such a desired property why shouldn't we just train tied SAEs?
>
> This is a very good question. We have performed the key experiment comparing the base, ours and tied methods in the Appendix D1. We discovered that the tied weights also reduce the number of dead neurons but lead to the worse reconstruction. Our method is the best of both words: achieves both the better reconstruction and reduces the number of dead neurons.

---

### Official Review · Reviewer_f19r · 2025-10-28

**Soundness:** 3
**Presentation:** 2
**Contribution:** 4
**Rating:** 6
**Confidence:** 4

**Summary:**

This paper makes the observation that for an SAE latent to be valid, the inner product between the encoder and decoder should be 1.0, and if it's not 1.0, then something is going wrong. The paper turns this into a metric called "alignment score", and shows that it matches the older and more expensive MCS score. The paper then further turns this observation into a constraint on the encoder, forcing it to always have an alignment score of 1.0. This training method is then validated in small LLM ReLU SAEs, and shows that it improves reconstruction and, as a side-effect, removes all dead latents.

**Strengths:**

- The observation that the dot product between and encoder and decoder latent should ideally be 1.0, and if it's not, something is wrong is a great observation that is obvious in hindsight.
- Turning this observation into a training constraint is also a great idea, as there's no reason to allow the SAE to learn clearly broken latents
- The resulting SAEs seem to be clearly superior to standard SAEs in the benchmarks shown
- This technique can also be applied to any untied SAE architecture, although this is not shown in the paper.
- The connection to the MCS score is also nice, as it gives further credibility to the metric

**Weaknesses:**

- The paper is written in a meandering way, and does not say up-front what the contributions are. I started reading the paper thinking it was about the MCS score and bi-modality, to discover that actually it's about this new SAE architecture that is much more exciting and interesting. This should be stated up-front! If you have created a new SAE architecture that seems to solve real problems with dead-latents and achieves good performance, you should lead with that!
- The paper only trains ReLU SAEs, even though the finding and technique should be applicable to any SAE architecture. BatchTopK SAEs are easy to train and state-of-the-art, these would be good to evaluate as well.
- The SAEs evaluated are only trained on 50M tokens and are very tiny (4k latents). It would be good to evaluate on an SAE size that might be used in reality, minimum 16k latents and 300M tokens.
- The experiments showing that the Aligned SAE achieves fewer dead latents than standard ReLU SAEs is not convincing. It should be very easy to train a ReLU SAE with 4k latents that has no dead latents. The authors should follow Conerly et al. 2024 (cited in the paper), and initialize the encoder to match the decoder with constant magnitude, and linearly increase the L1 penalty during the first 20% of training. If this is done, there should be not be dead latents.
- The SAEs trained are only evaluated on sparsity/reconstruction trade-off and number of dead features. It would be interesting to see if these SAEs are also better on downstream tasks like SAEBench.
- The paper does not compare aligned SAEs with tied SAEs or tied SAEs with unit-norm encoder/decoder, which should also achieve a perfect alignment score. I suspect that the SAEs in the paper will out-perform tied SAEs, but it would be good to confirm that the authors are not essentially re-inventing tied SAEs.
- It would be interesting to see more evaluations of how alignment score varies across SAE sizes / L0s, and different types of SAEs. You can probably just look at something like Gemma Scope SAEs and calculate alignment scores for all the pre-trained SAEs there (they have multiple sizes / L0 per layer).

**Questions:**

### Questions
- SAE latents where the encoder and decoder have 0 or even negative cosine similarity seem to be completely broken, and I would expect these to be dead latents. Did you verify that these latents are not dead?
- L178 "Naturally, this is a very artificial setting and the autoencoder can just learn the identity" - why can the SAE learn the identity here? This is a 1-latent SAE in the 2d space, it should not be possible to learn an identity matrix as the encoder and decoder are both rank 1
- Section 3.2: You could pretty easily turn the argument in this section into a proof, which would strengthen the paper. For instance, you could write a proof of this, put it in the appendix, and reference it at the end of the section.
- Why are aligned SAEs superior to a standard tied SAE where the decoder (and thus encoder) is forced to have norm 1? This would also achieve an alignment score of 1.0 and have even fewer hyperparams.
- It seems like this technique would also work just as well for any SAE architecture; why only evaluate on ReLU SAEs? ReLU SAEs are no longer used in practice. It would be ideal to evaluate this on JumpReLU or BatchTopK SAEs.
- Do you have any suspicions about why the SAEs don't always learn latents with alignment score of 1.0? I suspect that L1 SAEs in particular will suffer from shrinkage and this will harm their alignment score, while TopK/JumpReLU SAEs do not have this problem. I also wonder if feature hedging where the latents mix together correlated feature components might mess with the alignment score of latents in some way, either make it higher or lower than 1.0 slightly.
- How does this interact with feature absorption, where the SAE learns intentionally misaligned encoder and decoder directions? I suspect that even in this case the alignment score should be 1.0 here, but curious to hear your thoughts.
- How does a non-zero encoder bias interact with the idea that alignment score should be 1.0? Does this assume there's 0 encoder bias?

### Minor issues
- L31: "The proposed method..." Is this your proposed method or is this background info?
- L34-35: The phrasing "training an SAE **on** a dictionary of size N" is confusing. This should read "training an SAE **with** a dictionary of size N". The dictionary is a property of the SAE, not the training data of the SAE.
- nit: the paper uses backwards quotes at the start of quotations, e.g. L39, L28.
- nit: it seems like the paper is using the wrong latex citation keyword, making it hard to read. I think it should be `\citep` instead of `\citet`.
- L143: "It is not clear which of them is the true feature vector" I disagree with this statement, the decoder is the true feature. The decoder corresponds to the "dictionary" in dictionary learning theory and is what the SAE outputs to reconstruct the input. In dictionary learning theory there is typically not even an explicit encoder. The encoder only has to activate to the correct magnitude, but is otherwise free to project into the null space of the LLM without issue, and thus there are a potentially infinite number of vectors that are all equally valid encoders for a given decoder feature.

---

> ### Author Response · Authors · 2025-11-19
>
> We are deeply grateful for your time and valuable feedback, particularly for highlighting that the paper's narrative flow undersells its contributions. We agree that we should emphasize the invention of a new SAE type with improved performance and fewer dead features, and we appreciate your recognition of this discovery as exciting.
>
>
>
>
> ## 1. Improving TopK and BatchTopK SAEs
>
>
>
> > The paper only trains ReLU SAEs, even though the finding and technique should be applicable to any SAE architecture. BatchTopK SAEs are easy to train and state-of-the-art, these would be good to evaluate as well
>
>
>
> We agree. The new experiment added in Section 5.4 of the revised manuscript demonstrates that our method also improves modern architectures like TopK and BatchTopK, significantly strengthening our results.
>
> This addition further validates the robustness and applicability of our approach across diverse architectures.
>
>  ## 2. Small SAE runs
>
> >The SAEs evaluated are only trained on 50M tokens and are very tiny (4k latents). It would be good to evaluate on an SAE size that might be used in reality, minimum 16k latents and 300M tokens.
>
> We conducted experiments on dictionary sizes of 16K and 65K, demonstrating that our method reduces the number of dead neurons to almost none even in these more challenging scenarios. These results are now included in Appendix D5 of the revised manuscript.
>
> The choice of a 50M token budget was motivated by SAEBench [1], Section 4.4, which indicates that SAEs are mostly converged at this step. However, we also conducted a full 500M token training to directly compare our results with SAEs from SAEBench. These findings are detailed in Section 5.3 of the revised manuscript.
>
> ## 3. Other tricks for reducing the number dead features
>
> > The experiments showing that the Aligned SAE achieves fewer dead latents than standard ReLU SAEs is not convincing. It should be very easy to train a ReLU SAE with 4k latents that has no dead latents. The authors should follow Conerly et al. 2024 (cited in the paper), and initialize the encoder to match the decoder with constant magnitude, and linearly increase the L1 penalty during the first 20% of training. If this is done, there should be not be dead latents.
>
> We respectfully disagree. While the Conerly paper [2] claims their method reduces dead neurons to 1%, this was demonstrated in a very simplified setting using a one-layer transformer from "Towards Monosemanticity." We have already implemented their method of tied initialization and sparsity warmup, following the state-of-the-art repository suggested in the SAEBench paper.
>
> Even the SOTA SAEs trained by SAEBench with 4096 features exhibit a 20% dead latent rate, as listed by Neuronpedia statistics (which can be immediately confirm [here](https://www.neuronpedia.org/gemma-2-2b/12-sae_bench_0125-standard-res-4k__trainer_5)). This architecture, named standard_april_update, directly corresponds to the [2] training setting (often referred to as the Anthropic April Update).
>
> In other words, despite employing all available techniques, dead neurons remain a significant issue. The only methods we have found to address this without switching architectures are alignment training and weight tying (see the next answer).
>
> ## 4. Aligned training vs weight tying
>
> >The paper does not compare aligned SAEs with tied SAEs or tied SAEs with unit-norm encoder/decoder, which should also achieve a perfect alignment score. I suspect that the SAEs in the paper will out-perform tied SAEs, but it would be good to confirm that the authors are not essentially re-inventing tied SAEs.
>
> This is an important point, and we acknowledge that we should have included this ablation earlier. Interestingly, weight tying also reduces the number of dead neurons, but it comes at the cost of diminishing the overall quality of the SAE. We have added these experiments to the revised manuscript, with details in Appendix D1.
>
>  ## 5. Testing on the downstream task from SAEBench
>
> > The SAEs trained are only evaluated on sparsity/reconstruction trade-off and number of dead features. It would be interesting to see if these SAEs are also better on downstream tasks like SAEBench.
>
> Thank you for the remark. We tested the method for the downstream task of spurious correlation removal and have moved these results to Appendix D4 in the revised manuscript.
>
>
> [1] SAEBench: A Comprehensive Benchmark for Sparse Autoencoders in Language Model Interpretability
>
> [2] Update on dictionary learning improvements
>
> [3] Towards monosemanticity: Decomposing language models with dictionary

---

> > ### Author Response · Authors · 2025-11-19
> >
> > ## 6. The bias term
> >
> > > How does a non-zero encoder bias interact with the idea that alignment score should be 1.0? Does this assume there's 0 encoder bias?
> >
> > Why in the toy model scenario we used the zero bias, in the final experiment we do not use any simplifications like that. It is crucial, that the simplifications were only used as a motivation and the empirical results are realistic.
> >
> >  ## Minor
> >
> > > nit: the paper uses backwards quotes at the start of quotations, e.g. L39, L28.
> >
> > Thanks, fixed in the revised manuscript.
> >
> >
> >
> > >nit: it seems like the paper is using the wrong latex citation keyword, making it hard to read. I think it should be \citep instead of \citet.
> >
> > Thanks, fixed in the revised manuscript.
> >
> >
> >
> > >SAE latents where the encoder and decoder have 0 or even negative cosine similarity seem to be completely broken, and I would expect these to be dead latents. Did you verify that these latents are not dead?
> >
> > Interestingly, they are not always dead as presented in the scatter plot on the Figure 3.
> >
> >
> >
> >
> >
> >
> > **We hope the revisions and actions we've taken address your concerns. If there are any remaining issues or areas where further clarification or modifications could improve the paper, please let us know. We're committed to ensuring the quality and clarity of our work and would be happy to make additional adjustments.**

---

> ### Comment · Reviewer_f19r · 2025-11-25
>
> Thank you for the detailed response. I just wanted to follow up on this point:
>
> > Why in the toy model scenario we used the zero bias, in the final experiment we do not use any simplifications like that. It is crucial, that the simplifications were only used as a motivation and the empirical results are realistic.
>
> I think it is not completely correct to force the inner product of the encoder/decoder to 1.0 in the presence of encoder bias. If the inner product is forced to be 1.0, then it should only be able to reconstruct the input perfectly if the encoder bias is zero. If the encoder bias is negative, then the inner product will need to be slightly more than 1.0 to compensate, and if the encoder bias is positive, it will need to be slightly less than 1.0 to compensate. I thus suspect that by forcing the inner product to be 1.0 you are likely also forcing the encoder bias to be near 0.
>
> While I do not think this is a massive deal, I also suspect that allowing some leeway in your algorithm to account for the encoder bias would probably yield even better results. For instance, clamping the inner product to be between 0.9 and 1.1 (or something like that) would probably result in still keeping the encoder/decoder aligned (and thus avoiding dead latents) while allowing some flex to account for the encoder bias.

---

> > ### Author Response · Authors · 2025-11-25
> >
> > Thanks for the comment. It is an interesting point and idea which we can test experimentally.
> > On the other hand, we see the following issues with clamping:
> > - Switching from 1 to the interval $[1 - \epsilon, 1 + \epsilon]$ adds another hyperparameter. We really wanted to have a parameter-free method to avoid any additional tuning.
> > - More crucial issue is how to apply this clamping during training, doing it after every gradient update can lead to a unstable training and loss spikes. Our approach (formula 2) is just a simple projection
> >
> > In our simple toy model we on purpose used very crude assumptions and then relied on empirical validation. We believe that adding more details to a heuristic motivation would either:
> > - make it unfeasible
> > - make it unrealistic
> >
> > Thanks again for the idea, let us know if you have more questions. We will be glad to add farther clarification.

---

> > > ### Author Response · Authors · 2025-11-28
> > >
> > > Dear Reviewer f19r,
> > >
> > > I hope this message finds you well. With the discussion period drawing to a close in just a few days, we would like to thank you for the review. We have carefully considered your valuable comments and hope that we have effectively addressed all the remarks you raised.

---

### Official Review · Reviewer_uTqa · 2025-11-08

**Soundness:** 3
**Presentation:** 2
**Contribution:** 2
**Rating:** 4
**Confidence:** 4

**Summary:**

The paper investigates the critical issue of feature quality in sparse autoencoders (SAEs) for LLM interpretability, specifically the bimodal distribution of features into a group of useful, monosemantic features and a group of non-interpretable, artifact, or dead features. The authors present a computationally cheap alignment score ($W_{i,.}^{enc}\cdot W_{\cdot,i}^{dec}$) derived directly from the SAE weights as a superior proxy for feature quality compared to the costly maximum cosine similarity (MCS). They empirically confirm that this score is bimodal and highly correlated with MCS and auto-interpretable. Experiments show this aligned training method achieves a Pareto improvement in reconstruction metrics, drastically reduces dead features in a parameter-free manner, and yields better performance on a downstream bias removal task compared to standard training and state-of-the-art SAEBench models.

**Strengths:**

The paper successfully connects the known but computationally expensive bimodal phenomenon (observed via MCS) to a simple, elegantly derived alignment score that requires no additional data or training.
- The quality of the theoretical motivation is strong with a clear toy-model argument establishing that the ideal alignment score should be 1.0, which is then validated empirically with high correlation to established interpretability metrics like MCS and autointerpretability.
- The resulting novel training method is a major practical significance as it is an algebraic solution that fundamentally addresses a core limitation of SAEs, the emergence of artifact features leading to a parameter-free reduction of dead features and a Pareto improvement across reconstruction metrics on multiple models and dictionary sizes compared to both standard training and SOTA benchmarks.

**Weaknesses:**

- The primary weakness lies in the limited scope of the empirical validation and a slightly weak theoretical transition to the full model. While the paper establishes strong results for ReLU-based SAEs, the conclusion that the problem is fixed is premature as the new training method has not been extended to or tested on modern, non-ReLU architectures like TopK, Gated, or JumpReLU SAEs, where the bimodality is also shown to exist. This leaves the community uncertain about the generality and future relevance of the algebraic solution for non-ReLU contexts.

- The paper does not address a critical comparison, the need to test the aligned training method against other recently proposed non-standard training protocols (like square root, tanh, or p-annealing) that also aim to mitigate feature quality issues like feature shrinkage and dead features making it difficult to assess the distinct value or potential synergy of the aligned approach.
- The derivation for the ideal alignment score of 1.0 is based on a highly simplified, one-feature, one-data-point, no-sparsity-penalty toy model, and a more rigorous theoretical analysis or generalization explaining why this condition should hold in the high-dimensional, sparse, and multi-feature real-world setting would significantly strengthen the work.

**Questions:**

The authors should elaborate on their plans for extending the aligned training method to non-ReLU architectures such as TopK and Gated SAEs, as the presence of bimodality in these models is confirmed in Figure 1, yet the proposed algebraic solution appears specific to the ReLU encoder $f(x) := \text{ReLU}(W^{enc}x + b^{enc})$ structure.
- Follow-up question is to clarify the relationship between the proposed alignment score and the MCS score, given the correlation of r=0.65, what factors account for the remaining variance, and can features with high MCS and low alignment score (or vice-versa) provide new theoretical insights into feature quality or universality?
- Please provide an analysis or additional experiment comparing the aligned training method against other non-standard training methods designed to address feature issues, such as tanh penalty or p-annealing to determine if there is an overlap in their effect on dead feature reduction or if they can be combined for further performance gains.

---

> ### Author Response · Authors · 2025-11-19
>
> We are deeply grateful for your time and valuable input.
>
> We were pleased to hear that you found the theoretical motivation strong and that our solution is practical for effectively reducing the number of dead neurons. Your feedback has been instrumental in shaping the improvements we've made to the paper.
>
>
> ## 1. Modern SAE architectures
>
>
>
>
> > While the paper establishes strong results for ReLU-based SAEs, the conclusion that the problem is fixed is premature as the new training method has not been extended to or tested on modern, non-ReLU architectures like TopK, Gated, or JumpReLU SAEs, where the bimodality is also shown to exist. This leaves the community uncertain about the generality and future relevance of the algebraic solution for non-ReLU contexts.
>
>
>
> **Answer:** We acknowledge that this was a key limitation of our work. To address this, we have conducted additional experiments applying alignment training to TopK and BatchTopK architectures. Importantly, our method also improves reconstruction error and reduces the number of dead features in these modern architectures. These results are detailed in subsection 5.4 of the revised paper.
>
>
>
> ## 2. Combining alignment with p_anneal training
>
>
>
> >Please provide an analysis or additional experiment comparing the aligned training method against other non-standard training methods designed to address feature issues, such as tanh penalty or p-annealing to determine if there is an overlap in their effect on dead feature reduction or if they can be combined for further performance gains.
>
> We appreciate this feedback and have conducted an additional experiment, detailed in Section D.2, comparing the p_anneal approach to the combination of our alignment method with p_anneal. The results show that the combined approach achieves better reconstruction loss, while the reduction in dead neurons remains as significant as in the standard ReLU case. This demonstrates that our training improvement is not a reinvention of p_anneal. Instead, the synergy between the two methods suggests that the problems they address (bimodality and low-quality features (ours) versus feature shrinkage/suppression (p_anneal)) are distinct.
>
>
>
> ## 3. The alignment score vs MCS
>
>
>
> >Follow-up question is to clarify the relationship between the proposed alignment score and the MCS score, given the correlation of r=0.65, what factors account for the remaining variance, and can features with high MCS and low alignment score (or vice-versa) provide new theoretical insights into feature quality or universality?
>
> We intentionally do not place significant emphasis on the MCS score, as it was proposed by Sharkey et al. [1] as a rough approximation of feature quality in the absence of ground truth. While the correlation between the alignment score and MCS serves as a motivating argument for using the alignment score in training, our goal was never to perfectly explain MCS.
>
> As illustrated in the scatter plot in Figure 1, the occurrence of high MCS (>0.7) alongside low alignment is rare. Conversely, high alignment with low MCS is more common. This does not surprise us and happens even when a model is perfectly aligned (e.g., a SAE initialized with tied encoder and decoder) but still produces low-quality features. In essence, alignment is not a silver bullet but rather a method to enhance the training procedure and enforce the development of meaningful features.
>
>
>
>
>
> [1]  Taking features out of superposition with sparse autoencoders
>
>
>
> We appreciate your thoughtful feedback and hope these clarifications address your concerns. Thank you for your valuable insights!

---

> > ### Author Response · Authors · 2025-11-28
> >
> > Dear Reviewer uTqa,
> >
> > I hope this message finds you well. With the discussion period drawing to a close in just a few days, we would like to thank you for the review. We have carefully considered your valuable comments and hope that we have effectively addressed all the remarks you raised.

---

### Official Review · Reviewer_qySM · 2025-11-09

**Soundness:** 3
**Presentation:** 3
**Contribution:** 3
**Rating:** 6
**Confidence:** 2

**Summary:**

The paper studies the `bimodality` of feature quality in sparse autoencoders. It introduces (1) alignment score for each feature, given by the inner product between its encoder row and decoder column, and shows that this score is bimodal and correlates with MCS, autointerpretability, and dead features; and (2) an aligned-training parametrization that enforces alignment = 1 for every feature. On ReLU SAEs for Pythia and Gemma, aligned training largely removes dead features, improves or matches reconstruction metrics, and improves performance on a SAEBench spurious-correlation removal benchmark, outperforming SAEBench baselines.

**Strengths:**

- Simple, elegant alignment score that is cheap to compute and correlates well with existing quality metrics.

- Aligned-training parametrization is easy to implement, principled with respect to homogeneity, and substantially reduces dead features.

- Consistent gains or parity on reconstruction and improved performance on a SAEBench spurious-correlation task, including against strong baselines under matched compute.

- Good positioning in existing SAE literature and clear exposition of motivation and limitations.

**Weaknesses:**

- Aligned training is only implemented and evaluated for ReLU SAEs, despite bimodality also being shown for TopK/Gated/JumpReLU architectures.

- Interpretability evaluation is limited: mostly correlational (alignment vs autointerpretability) and lacks systematic autointerpretability comparisons or qualitative feature case studies.

- Robustness (multiple seeds, broader hyperparameter sweep, later layers / larger scales) and deeper theoretical explanation of why alignment reduces dead features are not fully developed.

**Questions:**

- Can you report autointerpretability metrics comparing aligned vs standard SAEs (not just correlations), and possibly include a few qualitative feature examples?

- Do you have initial results or a clear plan for extending aligned training to TopK/Gated/JumpReLU SAEs, and do you expect similar reductions in dead features there?

- How robust are the improvements across random seeds and different hyperparameters? Are there regimes where aligned training underperforms?

---

> ### Author Response · Authors · 2025-11-19
>
> We appreciate your valuable time and insightful comments. We are pleased to learn that you found our solution both simple and elegant.
>
> ## 1. Improving modern SAEs
>
> > Do you have initial results or a clear plan for extending aligned training to TopK/Gated/JumpReLU SAEs, and do you expect similar reductions in dead features there?
>
>
> **Answer:** We have performed a new experiment on the state-of-the-art architectures of TopK and BatchTopK sparse autoencoders. Notably, our approach enhances these models in terms of reconstruction loss and reduces the number of dead neurons. This experiment is detailed in subsection 5.4 of the revised paper, further solidifying our claims.
>
>
> ## 2. Correlation with autointerpretability
>
>
> > Can you report autointerpretability metrics comparing aligned vs standard SAEs (not just correlations), and possibly include a few qualitative feature examples?
>
>
>
> **Answer:**  The autointerpretability metric is known to be noisy, which is why we used it solely in a correlation experiment to motivate the alignment score as a feature quality proxy. We conducted the requested experiment, and the mean autointerpretability score of the aligned model trained with our method is slightly higher (0.271 vs. 0.26). Importantly, these scores are unaffected by dead neurons.
>
>
>
> ## 3. Robustness with respect to seed and hyperparameter.
>
> > How robust are the improvements across random seeds and different hyperparameters? Are there regimes where aligned training underperforms?
>
>
>
> **Answer:** The aligned training consistently improves performance across various dictionary sizes, as demonstrated in Appendix D5. Additionally, we conducted an experiment with three different random seeds, confirming that the effect of randomness is negligible (see Appendix D3).
>
>
>
>
> Thank you once again for taking the time to review the paper and for your valuable feedback!
>
> We hope the revisions and additional experiments we've included address your concerns.

---

> > ### Comment · Reviewer_qySM · 2025-11-26
> >
> > Thank you to the authors for the clarifications and additional experiments. I will keep my score as it is.

---

### Author Response · Authors · 2025-11-19

We have updated the revised manuscript to include the **crucial experiment on TopK and BatchTopK SAEs, addressing a main weakness highlighted by all reviewers** in section 5.4. This experiment demonstrates the effectiveness of our method across modern architectures, significantly strengthening our claims.

Moreover, to make our results reproducible, we have uploaded the code: https://anonymous.4open.science/r/sae_bimodality-ICLR/README.md

---

### Meta-Review · Area_Chair_K1a8 · 2026-01-06

**Summary:**

This paper identifies a bimodal distribution of features in Sparse Autoencoders (SAEs). Key contributions include an "alignment score", defined as the inner product between encoder and decoder weights, that serves as a proxy for feature quality. The authors introduces an "aligned-training" method that constrains this score to 1. The paper claims this achieves a Pareto improvement in reconstruction metrics and significantly reduces dead features across ReLU and modern architectures like TopK.

The proposed alignment score is computationally and conceptually simple, and the alignment training demonstrated empirical benefits. However, there are concerns on the following aspects:

-  The core motivation for the alignment constraint relies on a highly simplified toy model (one-feature, no-sparsity) and the authors did not sufficiently bridge the gap between this heuristic and the complexities of high-dimensional real-world settings.

- There remains skepticism regarding the "alignment" narrative. If alignment is the goal, then weight-tying is a more direct approach. However, the authors showed that weight-tying hurts reconstruction. The underlying mechanism for why the proposed "alignment" specifically succeeds where weight-tying fails is not deeply analyzed.

- Beyond reconstruction-sparsity tradeoffs, evidence that alignment directly leads to higher-quality features is limited

- Quality of presentation / figures.

Overall, the mechanistic interpretability community likely will find the empirical success of aligned training valuable. However, the paper may not entirely ready for publication and could benefit from working more along the lines above.

**Reviewer Concerns:**

Reviewer qySM

- Extending aligned training beyond ReLU SAEs: Addressed. The authors added Section 5.4, demonstrating that aligned training improves reconstruction and reduces dead neurons for TopK and Batch TopK SAEs.

- Limited interpretability evaluation: Addressed. Authors reported that the aligned model achieved a slightly higher mean autointerpretability score (0.271 vs 0.26).

- Robustness across seeds and the lack of a deep theoretical explanation for why alignment reduces dead features: Partially Addressed. Authors provided results for three different seeds (Appendix D3) and various dictionary sizes (Appendix D5).

Reviewer uTqa:

- Generality beyond ReLU: Addressed similar to that of qySM.

- Comparison with other training methods like p-annealing or tanh penalties: Addressed. Authors added Appendix D.2, showing that combining alignment with p-annealing yields better reconstruction loss than p-annealing alone.

- The toy model is too simplified to justify the alignment condition in high-dimensional settings: Not Addressed.


Reviewer f19r:

- The paper does not say up-front what the contributions are: Addressed. Authors agreed that they should highlight their contribution upfront.

- Generality beyond ReLU: Addressed similar to that of qySM.

- Evaluations on larger SAE sizes (16k+ latents) and longer training runs (300M+ tokens): Addressed. Authors added runs for 16K and 65K dictionaries (Appendix D5) and a full 500M token training run (Section 5.3).

- The authors should follow Conerly et al. 2024 to reduce dead latents: Addressed. Authors demonstrated that even SOTA SAEBench models using these tricks still exhibit significant dead latent rate.

- Testing on the downstream task from SAEBench: Addressed. Added to Appendex D1.

- Comparison with Tied SAEs: Addressed. Authors added Appendix D1, showing that while weight tying reduces dead neurons, it harms reconstruction quality, whereas their method improves both.

- More evaluations of how alignment score varies across SAE sizes / L0s, and different types of SAEs: Not Addressed.


Reviewer 72nk:

- Motivation of this work - should we just train tied SAEs and is there a conflict between the reconstruction objective and the alignment: Partially addressed. The author conducted experiments to show that tied SAEs hurts reconstruction quality. Meanwhile they highlighted that their method achieves the best of both world. However, the reviewer may have requested more analysis on why this is the case.

- Not convinced that the toy model is relevant for any practical case: Not addressed.

- No enough evidence that aligning features is correct or helpful: Partially addressed. The paper uses "the most common method in SAE literature which is by showing the better reconstruction-sparsity tradeoff"; however, it can be helpful to elaborate on why this is sufficient.

- Concern that the batch topk SAEs seem to be underperforming normal topk SAEs: Addressed. Authors argue based on evidence that there is no consistent difference between them.

**Reviewer Scores:**

Reviewer qySM:

Likely 6. The reviewer explicitly stated, "I will keep my score as it is" after reviewing.

Reviewer uTqa:

Likely 4. One of the main concern on toy model being too simplified is not addressed.

Reviewer f19r:

Likely 6 but may be higher given questions were mostly addressed.

Reviewer 72nk:

Likely 4 as many concerns are only partially addressed.

---

### Decision · Program_Chairs · 2026-01-26

Reject